# Effects of Mannan Oligosaccharides on Gas Emission, Protein and Energy Utilization, and Fasting Metabolism in Sheep

**DOI:** 10.3390/ani9100741

**Published:** 2019-09-28

**Authors:** Chen Zheng, Junjun Ma, Ting Liu, Bingdong Wei, Huaming Yang

**Affiliations:** 1College of Animal Science and Technology, Gansu Agricultural University, No. 1 Yingmen Village, Anning District, Lanzhou 730070, China; zhengc@gsau.edu.cn (C.Z.); majunjungsau@gmail.com (J.M.); 2Branch of Animal Husbandry, Jilin Academy of Agricultural Sciences, No.186 East Xinhua Road, Gongzhuling 136100, China; weibingdong@dlut.edu.cn (B.W.); yhmjl@163.com (H.Y.)

**Keywords:** energy, gas, mannan oligosaccharides, protein, sheep

## Abstract

**Simple Summary:**

Mannan oligosaccharides (MOS) are a promising feed additive to improve animal health, immune capacity, and antioxidation. Based on the previous studies, we carried out three experiments to investigate the effects of MOS on the gas emission, protein and energy utilization, and fasting metabolism of sheep. The results showed that 2.0% MOS supplementation led to the lowest *in vitro* CO_2_ production and lower CH_4_ production and decreased *in vivo* intake. However, it also decreased urine nitrogen excretion and energy released as CH_4_, and then improved the utilization of crude protein and energy of sheep. There were no differences in the parameters of respiration and energy metabolism of sheep under the fasting condition. The findings indicated that MOS slightly affected the gas emission and nutrients and energy utilization of sheep.

**Abstract:**

This study investigated the effects of mannan oligosaccharides (MOS) on *in vitro* and *in vivo* gas emission, utilization of crude protein (CP) and energy, and relative parameters of sheep under fasting metabolism conditions. *In vitro* gas productions were evaluated over 12 h in sheep diets containing different amounts of MOS (from 0% to 6.0%/kg, the increment was 0.5%). A control experiment was used to assess the gas emission, utilization of CP and energy, and fasting metabolism in control sheep and sheep treated with 2.0% MOS over 24 days (d). The results showed that 2.0% MOS supplementation led to the lowest *in vitro* CO_2_ production and less CH_4_ production, while also leading to decrease *in vivo* nutrients intake, CP and energy excretion, digested and retained CP, and energy released as CH_4_ (*p* < 0.05). Furthermore, 2.0% MOS supplementation appeared to decrease *in vivo* O_2_ consumption and CH_4_ production per metabolic body weight (BW^0.75^), and increase the CP retention rate of sheep (*p* < 0.074). MOS did not affect other parameters, along with the same parameters of sheep under fasting metabolism conditions (*p* > 0.05). The findings indicate MOS has only slight effects on the gas emission and nutrients and energy metabolism of sheep.

## 1. Introduction

Prebiotics are non-digestible food ingredients that benefit the host by selectively stimulating the activity of one or a limited number of bacteria in the intestine [1,2]. The predominant prebiotics include oligosaccharides such as fructooligosaccharides (FOS), inulin, mannan oligosaccharides (MOS), and xylooligosaccharides [1]. They have been termed a nutricine, meaning that they have no direct nutritive value, but maintain intestinal digestive and absorptive functions, thus improving the health and performance of farmed animals [3].

MOS are found in the outer layer of yeast, *Saccharomyces cerevisiae*, in the cell wall. As the most bioactive compounds, they contain both mannan proteins and complex carbohydrates, including β-glucans [4], and are widely used as a dietary supplement to boost the immune system and eliminate pathogens from the intestinal tract [5,6]. For instance, MOS bind to the mannose-specific lectin of Gram-negative pathogens that express Type-1 fimbriae (e.g., *Escherichia coli*), resulting in their excretion from the intestine [7]. The limited research focusing on MOS in ruminants found that MOS improved the immunoglobulin G (IgG) concentration of lamb’s blood [8], enhanced the health of the ruminal epithelium of sheep [9], improved the colostrum quantity of cows [4], and increased the antioxidant capacity of sheep [10]. In previous study, β 1-4 galacto-oligosaccharides (GOS) reduced the CH_4_ production in sheep [11]; however, very few scientists have studied the effect of MOS on the energy metabolism and gas emission of ruminants. So, we hypothesized that MOS could adjust the CO_2_ and CH_4_ emission in the rumen and improve the energy utilization of sheep. The purpose of this study was to investigate the effects of MOS on *in vitro* and *in vivo* gas production, respiration, protein digestion, energy metabolism, and fasting metabolism in sheep.

## 2. Materials and Methods

### 2.1. Experimental Design, Animals, and Housing

All experiments in this study were carried out in accordance with the approved guidelines of the Regulation of the Standing Committee of Gansu People’s Congress. All experimental protocols and the collection of samples were approved by the Ethics Committee of Gansu Agriculture University under permission no. DK-005.

A single-factor experimental design was used for the *in vitro* gas production experiments. Thirteen different doses of MOS (Bio-Mos^®^, Alltech, Nicholasville, KY, USA) were tested: 0, 0.5, 1.0, 1.5, 2.0, 2.5, 3.0, 3.5, 4.0, 4.5, 5.0, 5.5, and 6.0%/kg of basal diet (as fed basis). Each treatment was repeated three times. Four Chinese Northeast Merino rams with an eternal ruminal fistula and similar body weights (64.82 ± 3.17 kg) were used as ruminal fluid donors, and were fed basal diet and water ad libitum. An automatic recording device of trace gas production with six channels (Branch of Animal Husbandry, Jilin Academy of Agricultural Sciences, Gongzhuling, Jilin Province, China) equipped with gas flowmeters (CH_4_ and CO_2_, Sensors, Inc., Saline, MI, USA; H_2_, CITY Technology, Inc., Hampshire, UK) was used to measure the gas flow. The *in vitro* tests lasted for 12 h.

A control experiment was used to assess *in vivo* CO_2_, CH_4_, and NH_3_ production, O_2_ consumption, urine nitrogen excretion, total heat output, and the apparent digestibility and retention rates of crude protein (CP) and energy in control sheep and sheep treated with 2.0% MOS (because adding 2.0% MOS to the basal diet led to the lowest CO_2_ and lower CH_4_ production in an *in vitro* experiment; then, this dose was chosen to apply in an *in vivo* experiment). The experimental group consisted of eight healthy rams (*Dorper* ♂ × *Small tail Han-yang* ♀; four rams per group) with similar body weights (42.50 ± 3.24 kg). The test period included a 14-day (d) acclimation period, a 6-d digestive and metabolic experiment, a 60-hour (h) fasting acclimation period (the respiratory quotient (RQ) approximately was below 0.71 when sheep fasted for over 60 h), and a 24-h fasting metabolism experiment. All rams were housed individually in a respiratory chamber (Branch of Animal Husbandry, Jilin Academy of Agricultural Sciences, Gongzhuling, Jilin Province, China). Each chamber was equipped with a feeder and a drinker that provided ad libitum access to feed and water. Each chamber was also equipped with automatic gas flowmeters (CH_4_ and CO_2_, Sensors, Inc., Saline, MI, USA; O_2_, Advanced Micro Instruments, Inc., Costa Mesa, CA, USA; NH_3_, Industrial Health & Safety Instrumentation, Inc., St. Petersburg, FL, USA). Before rams went into the respiratory chamber, the 6.14-L calibrating CO_2_ was pumped into the respiratory chamber to test the recovery rate, and repeated three times; the result of the gas recovery tests was 98.22% (Appendix A). The data on gas production and O_2_ consumption were calibrated by the gas recovery rate of each chamber.

### 2.2. Experimental Diets

The basal diet was formulated to meet or exceed the recommendations for all nutrients of ram sheep under China Agricultural Industry Standard NY/T816-2004 (Table 1).

### 2.3. In Vitro Procedures

Rumen contents from each donor sheep were obtained immediately before morning feeding by soft tube and syringe via ruminal fistula and strained through four layers of cheesecloth. Then, the fluid from each sheep was pooled and mixed with culture medium solution in a 1:2 ratio (vol/vol) at 39 °C in accordance with the description by Menke et al. [14], and the mixed solution was also constantly filled with N_2_ to maintain anaerobic condition. Before incubation, in order to eliminate the metering error, 0.2843% calibrating CO_2_ was pumped into the channel, and then the gas flowmeter was calibrated to 0.2843%. All *in vitro* cultures contained 2 g of experimental diets, which were carefully weighed into the fermentation channel of the automatic recording device of trace gas production, while the control channel did not receive any diet in order to calibrate the gas production from the ruminal fluid. Additionally, different doses of MOS were added into the channel in terms of experimental treatment. Then, each channel received 150 mL of mixed solution (including 50 mL of ruminal fluid and 100 mL of culture medium solution), and was maintained at 39 °C for 12 h of incubating. Since there were six channels for incubation, the culture was divided into nine periods, which meant that control, 0%, 0.5%, 1.0%, 1.5%, and 2.0% MOS treatments were carried out during the first period; control, 2.5%, 3.0%, 3.5%, 4.0%, and 4.5% MOS treatments were carried out during the second period; control, 5.0%, 5.5%, and 6.0% MOS treatments were carried out during the third period; and then, this procedure was repeated two times during the fourth to ninth periods. The productions of CO_2_, CH_4_, and H_2_ were recorded by a gas flowmeter automatically per every 6 min. Then, the gas production was calculated per min from a total of 12 h of incubated gas production, after which the gas production per every 24 h was calculated.

### 2.4. In Vivo Procedures

In the acclimation period and digestive and metabolic experiments, all sheep received a basal diet or 2.0% MOS-treated diet randomly. Since they were housed in the respiratory chamber, the productions of CO_2_, CH_4_, NH_3_, and H_2_ along with the consumption of O_2_ were measured using the gas flowmeter automatically per every 27 min.

The weight of feed intake and residues were recorded daily carefully. During the digestive and metabolic experiments, 10% of total diet, 10% of total feces output, and 5% of total urine output (5 mL of sulfuric acid added into total urine daily before collection to prevent nitrogen losses) were sampled for 6 sequential d and stored at −20 °C. At the end of the data collection period, the diets, fecal samples, and urine samples were thawed and pooled for each sheep. For nitrogen analysis, 3% of total feces output was sampled daily and stored in wide-mouth bottles containing 20 mL of 10% sulfuric acid for nitrogen fixation, and these samples were also pooled for 6 d individually.

After the digestive and metabolic experiments, sheep fasted approximately over 60 h until the RQ was below 0.71. Then, the sheep fasting metabolism experiment was carried out. The procedures of gas production and O_2_ consumption recording, feces output sampling, and urine output sampling was as previously stated, but the recording and sampling times were 24 h.

### 2.5. Chemical Analysis

The diets and fecal samples were dried at 65 °C for 72 h in a forced-air oven, and then ground through a 1-mm screen using a Wiley Mill (Ogaw Seiki Co., Ltd., Tokyo, Japan). The gross energy of diet, feces, and urine samples were measured using a calorimeter (C2000, IKA^®^-Werke GmbH & Co. KG, Staufen, Germany). The nitrogen contents of the diet, feces containing 10% sulfuric acid, and urine samples were determined by the Kjeldahl method (AOAC, method 990.03) [12].

### 2.6. Calculations and Statistical Analysis

A gas flowmeter records the dynamic gas production or consumption; therefore, the total gas production or consumption is calculated using the following equations:*In vitro* gas production (mL/24 h) = average production (mL/min) × 60 × 24;
*In vivo* gas production/consumption (L/24 h) = average production/consumption (L/min) × 60 × 24.

Parameters related to energy metabolism are calculated using the following equations [15]:
Total heat output (J) = 16.175 × O_2_ (L) + 5.021 × CO_2_ (L) − 2.167 × CH_4_ (L) − 5.987 × Urine nitrogen (g).
Oxidation of protein (g) = Urine nitrogen (g) × 6.25.
CO_2_ production from the oxidation of protein (L) = Urine nitrogen (g) × 4.754.
O_2_ consumption from the oxidation of protein (L) = Urine nitrogen (g) × 5.923.
Heat output from the oxidation of protein (J) = Urine nitrogen (g) × 113.76.

Data from the *in vitro* experiment were analyzed by one-way analysis of variance (SPSS 19.0, IBM Co. Limited, Chicago, IL, USA) using the following model:*X_ij_* = *μ* + *α_i_* + *e_ij_*(1)
where *X_ij_* is the observation of the dependent variable (*i* = 1 to 13, *j* = 1 to 3), *μ* is the population mean, *α_i_* is the random effect of treatment, and *e_ij_* is the random error associated with the observation.

Data from the *in vivo* experiments were analyzed by an independent samples *t* test.

Significance was determined at *p* ≤ 0.05 and tendency at 0.05 < *p* ≤ 0.10 using Tukey’s multiple comparison tests.

## 3. Results

### 3.1. In Vitro CO_2_, CH_4_ and H_2_ Production

The 24-h *in vitro* CO_2_, CH_4_, and H_2_ production under different amounts of adding MOS are shown in Table 2. MOS influenced *in vitro* CO_2_ and CH_4_ productions significantly (*p* < 0.05). An MOS of 3.5% led to the highest production of CO_2_ and CH_4_, while 2.0% MOS resulted in the lowest amount of CO_2_ emissions, and 0.5% and 5.0% MOS resulted in the lowest CH_4_ emissions (*p* < 0.05). No differences in H_2_ production were observed between different MOS doses (*p* > 0.05).

### 3.2. In Vivo Gas Production, CP, and Energy Digestion and Retention of Sheep

The CO_2_, CH_4_, and NH_3_ production, O_2_ consumption, urine nitrogen excretion, total heat output, and relative parameters regarding the energy metabolism of the control and 2.0% MOS-treated sheep are shown in Table 3. MOS did not affect RQ, CO_2_, CH_4_, and NH_3_ production, O_2_ consumption, and total heat output (*p* > 0.05). However, MOS decreased the urine nitrogen excretion and parameters of protein oxidation related to the energy metabolism of sheep (*p* < 0.05). Furthermore, MOS tended to decrease O_2_ consumption and CH_4_ production per metabolic body weight (BW^0.75^), along with the ratio of heat output from protein oxidation to total heat output (*p* < 0.074).

The effects of MOS on sheep dry matter intake (DMI), CP, and energy digestion and retention are shown in Table 4. The supplementation of MOS decreased dry matter (DM), CP, and energy intake, CP and energy excreted in feces and urine, digested and retained CP, and the energy released as CH_4_ in sheep (*p* < 0.05). The addition of MOS did not affect the apparent digestibility of CP and energy, retention of energy, digestible energy (DE), and metabolizable energy (ME) (*p* > 0.05). However, MOS tended to increase the retention rate of CP (*p* < 0.054).

### 3.3. In Vivo Gas Production and Energy Metabolism of Sheep under Fasting Metabolism Conditions

There were no differences in RQ, CO_2_ production, O_2_ consumption, urine nitrogen excretion, total heat output, and relative parameters regarding energy metabolism between the control and MOS-treated sheep under fasting metabolism conditions (*p* > 0.05, Table 5).

## 4. Discussion

### 4.1. In Vitro CO_2_, CH_4_ and H_2_ Production

In this study, the MOS addition range from 0.5% to 2.0% decreased *in vitro* CO_2_ and CH_4_ production. Basically, carbohydrate was degraded by microbes in the rumen to produce approximately 65.5% CO_2_, 28.8% CH_4_, and small quantities of N_2_, O_2_, and H_2_ [16]. Although some bicarbonates from saliva through the ruminal wall produce CO_2_, the degradation of carbohydrates by ruminal microbes is the major pathway for CO_2_ production [16]. In the rumen, methanogenesis evolves from carbon oxide, acetic acids, and methanol by methanogens. For most methanogens, methanogenesis from CO_2_ and H_2_ is the sole energy source. CH_4_ production is considerably enhanced by the CO_2_ present in the headspace, and there is an equilibrium established between the dissolved CO_2_ in the media and the partial pressure of CO_2_ in headspace gas. Meanwhile, the higher CO_2_ concentration in the headspace would result in a greater dissolved CO_2_ concentration in the media [17,18]. The result of the current study was in accordance with the previous conclusion that a positive relationship existed between CO_2_ and CH_4_; however, some scientists believed that there was a negative relationship, because CO_2_ and H_2_ are in general the precursors for CH_4_ formation in the rumen [19,20]. This experiment illustrated that in an *in vitro* condition, MOS as an additive had only slight effects on ruminal fermentation, the partial pressure of CO_2_, and methanogens and methanogenesis, and adding 0.5–2.0% MOS decreased CO_2_ production and led to lower CH_4_ production. It might be that MOS doses ranging from 0.5% to 2.0% were appropriate doses for ruminal microbes’ fermentation pathway, and most of the produced CO_2_ was potentially absorbed by the rumen wall to synthesize some nutrients, such as aspartic acid and isoleucine, and a lower partial pressure of CO_2_ and dissolved CO_2_ led to lower methanogenesis. Furthermore, higher doses of MOS meant higher oligosaccharides transferred into rumen as a substrate for microbes’ fermentation, so, more substrates produced more CO_2_, and instead promoted methanogenesis. However, MOS doses ranging from 2.5% to 4.5% generated the highest CO_2_ and CH_4_ emissions, but MOS doses ranging 5.0% to 6.0% resulted in lower CO_2_ and CH_4_ production. Further investigation is needed to confirm whether moderate doses of MOS improve carbohydrate fermentation in the rumen and much higher doses of MOS inhibit ruminal fermentation because more soluble carbohydrates led to lower pH and were harmful to microbes [20]. In addition, the fate of CO_2_ is more complicated because of the pooling and recycling of animal metabolic carbon as urea and bicarbonates in saliva with that produced by the rumen organisms [21]. Furthermore, the process of methanogenesis is affected by many environmental factors, including the carbohydrate type and digestion passage rate. Feed additives such as yeast have the ability to shift H_2_ utilization from methanogenesis to reductive acetogenesis through the homoacetogenic bacteria that can produce acetate from CO_2_ and H_2_ [22], along with internal factors [23]. So, more research about MOS on ruminal fermentation and the microbial population need to be undertaken in the future.

### 4.2. In Vivo Gas Production, CP, and Energy Digestion and Retention of Sheep

In the present study, the supplementation of MOS decreased ingestion, CP, and energy excreted in feces and urine, digested and retained CP, energy released as CH_4_, along with the same parameters per BW^0.75^, but MOS did not affect the apparent digestibility of CP and energy. Surprisingly, MOS decreased the ingestion of DM in sheep; the reason may be that there were low replicates in each treatment, indicating that individual differences may be a key factor in the results. Additional experiments with more sheep are needed to verify these effects of MOS.

Although MOS decreased the DMI of sheep, it decreased the CP and energy excretion in feces and urine at the same time; as a result, no differences were observed on the digestion and retention of CP and energy, even though higher digestibility and retention rates occurred in MOS-treated sheep. It indicates that MOS negatively influences the nutrients and energy intake of sheep; however, it makes sheep improve the utilization of nutrients and energy. Accordingly, because of improving the utilization of energy both in the rumen and the whole body of sheep, the methanogenesis in rumen was restricted, and resulted in less energy released as CH_4_. A previous study found that GOS significantly increased the DE of sheep fed with Italian ryegrass hay and concentrate (3:2, on a DM basis), but did not affect nitrogen digestion and retention [24]. Other researchers reported that MOS did not affect the apparent digestibility and retention rate of nutrients in cattle and sheep [4,10]; these findings are in accordance with that of Goiri et al. [25], who pointed out that chitosan did not influence the nutrients’ apparent digestibility. However, another study indicated that MOS enhanced the health of the ruminal epithelium of sheep by reducing the thickness of the stratum corneum, and it might have increased the nutrients digestion [9]. In addition, similar to this study, Sharma et al. [26] also reported that MOS increased the apparent digestibility of CP in Murrah buffalo calves. These findings indicated that MOS improved the CP utilization via decreased urine nitrogen excretion. So, more research about the effects of MOS on nutrients and energy metabolism in rumen also need to be undertaken in the future.

In the current study, treatment with 2.0% MOS resulted in less DM ingestion compared to the control. This led to less O_2_ consumption and CO_2_ production, because the digestion of feed and metabolism of nutrients in MOS-treated sheep consumed less O_2_ and produced less CO_2_. In *in vivo* conditions, CO_2_ production from respiration is much larger than ruminal fermentation, and MOS only adjusts ruminal fermentation slightly; so, the decreasing tendency of CH_4_ is observed from sheep in the respiratory chamber, not CO_2_ production. Thus, it is possible that the basic vital activities of sheep were not affected by MOS. Respiration is a kind of rhythmic vital activity; its frequency and amplitude are determined by the body condition, which depends on nervous humoral regulation. Normally, feed intake cannot change the basic respiration frequency and amplitude, not to mention that there was only a small quantity of MOS; the results of the current study are also in consonance with the physiological theory.

Food ingestion affects the production of body heat in animals; however, for example, animals continuously produce heat and lose it to their surroundings, either directly by radiation, conduction, and convection, or indirectly by the evaporation of water [27]. In the current study, MOS decreased the urine nitrogen excretion and relative protein oxidation in sheep; however, the ratio of heat output from protein oxidation in the body to total heat output was decreased at the same time, indicated that the basal metabolism process is hardly influenced by MOS.

### 4.3. In Vivo Gas Production and Energy Metabolism of Sheep under Fasting Metabolism Conditions

MOS did not affect the CO_2_ production, O_2_ consumption, urine nitrogen excretion, total heat output, and relative parameters regarding the energy metabolism of sheep under fasting metabolism conditions in the current study. In a fasting animal, the quantity of heat produced is equal to the energy of the tissue catabolized. When measured under specific conditions, the energy is known as the animal’s basal metabolism. A fasting animal must oxidize reserves of nutrients to provide the energy needed for essential processes such as respiration and circulation of the blood [27]. MOS cannot modify primary vital activities; no differences were observed in CO_2_ production, O_2_ consumption, urine nitrogen excretion, total heat output, and relative parameters regarding energy metabolism in 2.0% MOS-treated sheep and non-MOS fed sheep under fasting metabolism conditions.

## 5. Conclusions

The supplementation of MOS did not affect *in vitro* H_2_ production, *in vivo* CO_2_, CH_4_, and NH_3_ production, O_2_ consumption, total heat output, the apparent digestibility of CP and energy, the retention rates of energy of sheep, and the respiration and energy metabolism of sheep under fasting metabolism conditions. However, the addition of 2.0% MOS to the sheep diet led to the lowest *in vitro* CO_2_ production and less CH_4_ production. Furthermore, treatment with 2.0% MOS decreased the intake of DM, CP, and energy, along with the CP and energy in feces and urine, digested and retained CP, and released the energy as CH_4_. Treatment with 2.0% MOS appeared to increase the retention rate of CP in sheep. The results suggest that MOS only slightly affects ruminal fermentation and metabolism in sheep, and the effects were not strong enough to lead to substantial changes.

## Figures and Tables

**Table 1 animals-09-00741-t001:** Ingredient and chemical composition of the basal diet.

Items	Concentrate
Ingredient (%)	
Corn	24.90
Soybean meal	12.72
Chinese wildrye	60.00
Calcium hydrophosphate	0.78
Limestone	0.80
Salt	0.40
Additive premix ^1^	0.40
Chemical composition (%) ^2^	
DM	85.71
DE (MJ/kg) ^3^	9.25
CP	9.46
DE/CP (MJ/g)	0.10
NDF	46.06
ADF	23.75
Ca	0.65
P	0.40

^1^ Additive premix includes mineral elements (mg/kg): S, 200; Fe, 25; Zn, 40; Cu, 8; I, 0.3; Mn, 40; Se, 0.2; Co, 0.1; vitamins (IU/kg): vitamin A, 940; vitamin E, 20. ^2^ Concentrations of dry matter (DM), crude protein (CP), Ca, and P were measured in accordance with AOAC (DM: method 930.15, CP: method 990.03, Ca: method 978.02, P: method 946.06) [12], and concentrations of neutral detergent fiber (NDF) and acid detergent fiber (ADF) were measured in accordance with Goering and Soest [13]. ^3^ DE = digestible energy.

**Table 2 animals-09-00741-t002:** *In vitro* CO_2_, CH_4_, and H_2_ production with different amounts of adding mannan oligosaccharides (MOS).

MOS (%) ^1^	CO_2_ (mL/24 h)	CH_4_ (mL/24 h)	H_2_ (mL/24 h)
0	161.52 ^cde^	27.87 ^bc^	0.11
0.5	153.03 ^cde^	26.21 ^c^	0.06
1.0	155.61 ^cde^	28.29 ^bc^	0.04
1.5	145.64 ^de^	28.50 ^bc^	0.10
2.0	143.65 ^e^	27.65 ^bc^	0.10
2.5	169.92 ^abc^	30.80 ^abc^	0.05
3.0	169.16 ^abc^	30.88 ^abc^	0.06
3.5	189.27 ^a^	36.58 ^a^	0.08
4.0	186.53 ^ab^	34.66 ^ab^	0.09
4.5	172.96 ^abc^	28.92 ^abc^	0.11
5.0	162.40 ^cde^	26.64 ^c^	0.10
5.5	166.69 ^bcd^	27.73 ^bc^	0.09
6.0	162.60 ^cde^	27.12 ^bc^	0.10
SEM ^2^	2.05	0.58	0.01
*p*-value	<0.001	0.001	0.841

^1^ MOS doses were 0, 0.5, 1.0, 1.5, 2.0, 2.5, 3.0, 3.5, 4.0, 4.5, 5.0, 5.5, and 6.0% added to a 2-g basal diet. Mean results of *in vitro* gas production are shown for yield per 24 h (*n* = 3 per treatment, *in vitro* gas production (mL/24 h) = average production (mL/min) × 60 × 24). ^2^ SEM: standard error of mean. Values within a column with different superscripts differ significantly at *p* < 0.05.

**Table 3 animals-09-00741-t003:** *In vivo* gas production and energy metabolism of control and MOS-treated sheep.

Item ^1^		Control	MOS-Treated	SEM ^9^	*p*-Value
RQ ^2^		0.86	0.87	0.002	0.328
CO_2_ production	(L/24 h)	375.05	372.29	4.04	0.111
	(L/24 h/kg BW^0.75^) ^3^	20.77	20.58	0.20	0.641
CH_4_ production	(L/24 h)	21.64	20.22	0.40	0.600
	(L/24 h/kg BW^0.75^)	1.20	1.12	0.02	0.072
NH_3_ production	(L/24 h)	32.58	33.43	0.61	0.923
	(L/24 h/kg BW^0.75^)	1.80	1.85	0.03	0.507
O_2_ consumption	(L/24 h)	438.37	430.69	4.96	0.074
	(L/24 h/kg BW^0.75^)	24.27	23.81	0.25	0.358
Urine nitrogen excretion	(g/24 h)	14.87 ^a^	13.56 ^b^	0.30	0.028
	(g/24 h/kg BW^0.75^)	0.82 ^a^	0.75 ^b^	0.02	0.035
Total heat output ^4^	(MJ/24 h)	8.65	8.57	0.10	0.694
	(MJ/24 h/kg BW^0.75^)	0.48	0.47	0.01	0.553
Oxidation of protein ^5^	(g/24 h)	92.93 ^s^	84.78 ^b^	1.88	0.028
	(g/24 h/kg BW^0.75^)	5.15 ^a^	4.70 ^b^	0.11	0.035
CO_2_ production from the oxidation of protein ^6^	(L/24 h)	70.69 ^a^	64.48 ^b^	1.43	0.028
	(L/24 h/kg BW^0.75^)	3.92 ^a^	3.57 ^b^	0.08	0.035
O_2_ consumption from the oxidation of protein ^7^	(L/24 h)	88.07 ^a^	80.34 ^b^	1.78	0.028
	(L/24 h/kg BW^0.75^)	4.88 ^a^	4.45 ^b^	0.10	0.035
Heat output form the oxidation of protein ^8^	(MJ/24 h)	1.69 ^a^	1.54 ^b^	0.03	0.028
	(MJ/24 h/kg BW^0.75^)	0.094 ^a^	0.086 ^b^	0.002	0.035
The ratio of heat output from the oxidation of protein to total heat output (%)		19.55	18.12	0.39	0.069

^1^ Sheep were fed diets containing 0% and 2.0% MOS (*n* = 4 per treatment). Mean results of CO_2_, CH_4_, and NH_3_ production, O_2_ consumption, urine nitrogen excretion, total heat output, and relative parameters regarding energy metabolism are shown for the 6-d collection phase of the study for each treatment. The *in vivo* gas production or consumption was calculated using the following equation: *in vivo* gas production/consumption (L/24 h) = average production/consumption (L/min) × 60 × 24. ^2^ RQ = respiratory quotient. ^3^ BW^0.75^ = Metabolic body weight. ^4^ Total heat output (J) = 16.175 × O_2_ (L) + 5.021 × CO_2_ (L) − 2.167 × CH_4_ (L) − 5.987 × Urine nitrogen (g). ^5^ Oxidation of protein (g) = Urine nitrogen (g) × 6.25. ^6^ CO_2_ production from the oxidation of protein (L) = Urine nitrogen (g) × 4.754. ^7^ O_2_ consumption from the oxidation of protein (L) = Urine nitrogen (g) × 5.923. ^8^ Heat output from the oxidation of protein (J) = Urine nitrogen (g) × 113.76. ^9^ SEM: standard error of the mean. Values within a row with different superscripts differ significantly at *p* < 0.05.

**Table 4 animals-09-00741-t004:** *In vivo* nutrients and energy digestion and retention of control and MOS-treated sheep.

Item ^1^		Control	MOS-Treated	SEM ^11^	*p*-Value
Dry matter intake (DMI, kg/24 h)		1.66 ^a^	1.48 ^b^	0.02	<0.001
Crude protein (CP)					
CP intake	(g/24 h)	182.96 ^a^	163.56 ^b^	2.27	<0.001
	(g/24 h/kg BW ^0.75^) ^2^	10.07 ^a^	9.22 ^b^	0.13	<0.001
CP in feces	(g/24 h)	46.90 ^a^	38.78 ^b^	2.03	0.044
	(g/24 h/kg BW ^0.75^)	2.51	2.12	0.12	0.107
CP in urine	(g/24 h)	92.93 ^a^	84.78 ^b^	1.88	0.028
	(g/24 h/kg BW ^0.75^)	9.40	9.41	0.19	0.976
Digested CP ^3^	(g/24 h)	136.05 ^a^	124.78 ^b^	2.14	0.007
	(g/24 h/kg BW ^0.75^)	7.56	7.09	0.12	0.058
Retained CP ^4^	(g/24 h)	43.12 ^a^	40.00 ^b^	0.66	0.016
	(g/24 h/kg BW ^0.75^)	2.34	2.28	0.03	0.339
Apparent digestibility (%) ^5^		74.41	76.40	1.02	0.336
Retention rate (%) ^6^		23.56	24.42	0.22	0.054
Energy					
Energy intake	(MJ/24 h)	35.39 ^a^	31.64 ^b^	0.44	< 0.001
	(MJ/d/kg BW ^0.75^)	1.96 ^a^	1.73 ^b^	0.03	< 0.001
Energy in feces	(MJ/24 h)	12.69 ^a^	10.05 ^b^	0.61	0.028
	(MJ/d/kg BW ^0.75^)	0.74 ^a^	0.48 ^b^	0.03	<0.001
Energy in urine	(MJ/24 h)	0.46 ^a^	0.42 ^b^	0.01	0.028
	(MJ/d/kg BW ^0.75^)	0.03	0.02	0.001	0.301
Energy in CH_4_	(MJ/24 h)	0.86 ^a^	0.80 ^b^	0.01	0.015
	(MJ/d/kg BW ^0.75^)	0.05	0.04	0.001	0.094
Digestible energy (DE) ^7^	(MJ/24 h)	22.70	21.59	0.46	0.231
	(MJ/d/kg BW ^0.75^)	1.22	1.25	0.02	0.549
Metabolizable energy (ME) ^8^	(MJ/24 h)	21.40	20.39	0.45	0.270
	(MJ/d/kg BW ^0.75^)	1.15	1.18	0.02	0.470
Apparent digestibility (%) ^9^		64.34	68.59	1.52	0.165
Retention rate (%) ^10^		60.63	64.32	1.48	0.216

^1^ Sheep were fed 0% and 2.0% MOS (*n* = 4 per treatment). Mean results of DMI, CP, and energy digestion and retention are shown for the 6-d collection phase of the study for each treatment. ^2^ BW ^0.75^ = Metabolic body weight. ^3^ Digested CP (g) = CP intake − CP in feces. ^4^ Retained CP (g) = CP intake − CP in feces − CP in urine. ^5^ CP apparent digestibility (%) = (CP intake − CP in feces)/(CP intake). ^6^ CP retention rate (%) = (CP intake − CP in feces − CP in urine)/(CP intake). ^7^ Digestible energy (MJ) = energy intake − energy in feces. ^8^ Metabolizable energy (MJ) = energy intake − energy in feces − energy in urine – energy in CH_4_. ^9^ Energy apparent digestibility (%) = (energy intake – energy in feces)/(energy intake). ^10^ Energy retention rate (%) = (energy intake − energy in feces − energy in urine − energy in CH_4_) / (energy intake). ^11^ SEM: standard error of the mean. Values within a row with different superscripts differ significantly at *p* < 0.05.

**Table 5 animals-09-00741-t005:** *In vivo* gas production and energy metabolism of control and MOS-treated sheep under fasting metabolism conditions.

Item ^1^		Control	MOS-treated	SEM ^9^	*p*-Value
RQ ^2^		0.66	0.69	0.006	0.267
CO_2_ production	(L/24 h)	69.00	75.03	3.93	0.849
	(L/24 h/kg BW^0.75^) ^3^	3.83	4.15	0.22	0.494
O_2_ consumption	(L/24 h)	103.45	109.57	5.24	0.921
	(L/24 h/kg BW^0.75^)	5.74	6.06	0.29	0.617
Urine nitrogen excretion	(g/24 h)	3.28	2.95	1.52	0.828
	(g/24 h/kg BW^0.75^)	0.18	0.35	0.08	0.490
Total heat output ^4^	(MJ/24 h)	2.03	1.99	0.18	0.783
	(MJ/24 h/kg BW^0.75^)	0.11	0.11	0.004	0.788
Oxidation of protein ^5^	(g/24 h)	20.53	18.42	3.87	0.828
	(g/24 h/kg BW^0.75^)	1.26	1.11	0.26	0.824
CO_2_ production from the oxidation of protein ^6^	(L/24 h)	15.61	14.01	2.94	0.828
	(L/24 h/kg BW^0.75^)	0.96	0.85	0.20	0.824
O_2_ consumption from the oxidation of protein ^7^	(L/24 h)	19.45	17.45	3.67	0.828
	(L/24 h/kg BW^0.75^)	1.19	1.05	0.24	0.824
Heat output form the oxidation of protein ^8^	(MJ/24 h)	0.37	0.34	0.07	0.828
	(MJ/24 h/kg BW^0.75^)	0.02	0.02	0.005	0.824
The ratio of heat output from the oxidation of protein to total heat output (%)		19.84	19.09	4.95	0.953

^1^ Sheep were under the fasting metabolism conditions (*n* = 4 per treatment). Mean results of CO_2_ production, O_2_ consumption, urine nitrogen excretion, total heat output, and relative parameters regarding the energy metabolism of control and MOS-treated sheep are shown for the 24-h collection phase of the study for each treatment. The *in vivo* gas production or consumption was calculated using following equation: *in vivo* gas production/consumption (L/24 h) = average production/consumption (L/min) × 60 × 24. ^2^ RQ = respiratory quotient. ^3^ BW^0.75^ = Metabolic body weight. ^4^ Total heat (J) = 16.175 × O_2_ (L) + 5.021 × CO_2_ (L) − 5.987 × Urine N (g). ^5^ Oxidation of protein (g) = Urine nitrogen (g) × 6.25. ^6^ CO_2_ production from the oxidation of protein (L) = Urine nitrogen (g) × 4.754. ^7^ O_2_ consumption from the oxidation of protein (L) = Urine nitrogen (g) × 5.923. ^8^ Heat output from the oxidation of protein (J) = Urine nitrogen (g) × 113.76. ^9^ SEM: standard error of the mean.

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
