# Peer review of "Effects of Mannan Oligosaccharides on Gas Emission, Protein and Energy Utilization, and Fasting Metabolism in Sheep"

_animals, 2019, doi:10.3390/ani9100741_

Round 1
Reviewer 1 Report
In my opinion the Manuscript titled “Effects of mannan oligosaccharides on gas emission, protein and energy utilization, and fasting metabolism in sheep” (animals-585569) is an original scientific paper in the broad areas of Animals journal. The subject (effects of MOS on in vitro and in vivo gas emission, utilization of crude protein and energy, and relative parameters in sheep under fasting metabolism conditions) is a topic interesting to look into.
I think the manuscript needs some revisions before publication.
In general, I think that the manuscript is well written respecting the Animals journal indications.
Keywords: I prefer use words not present in the title
Material and methods: The experimental design is complex due to the in vivo and in vitro part. However, some details should be added, for example amount of medium in the fermentation channel in the in vitro trial, how the conversion of gas for 12h to 24h was made and some specification for the chemical analyses of basal diet (i.e. NDF, ADF, Ca, P reported in table 1). Probably, more references for protocol used needs to be inserted.
Results: can be improved (see specific suggestions).
Discussion: the results obtained were well discussed and justified; the data have been adequately commented and compared with similar studies.
Specific suggestions:
P1L17: move ‘in vitro’ after ‘led’
P1L32-33: the sentence is not clear please re-write
P2L72-73: the sentence is not clear please clarify
PL103: normally a flow of CO2 is used…
P4L134-135: not clear ‘…for energy measurements…’
P4L1557-156: change the sentence as follows ‘where X is the observation of the dependent variable (i = 1 to 13), μ is the population mean, α is the random effect of treatment, and e is the random error associated with the observation.’
P4L157-161: delete this part, it is not necessary
P4-5L166-174: all this part should be re-written, because it is not easy to understand. The statistical significances are already reported in the table…, you should just underline the highest or lowest values
P5L181-182: simplify the title… (also elsewhere)
P8L243: delete ‘approximately’
P10L324-327: improve the grammar of the sentence (after semicolon there is no verb)
Tables:
Table 2: It is not clear why and how this was made ‘Mean results of in vitro gas production are shown for 12-h collection phase of the study for each treatment (n = 3 per treatment), and then converted into yield per 24-h (in vitro gas production (mL/24 h) = average production (mL/min) × 60 × 24).’
In the legend delete a,b before values.
Tables 3, 4 and 5: The titles are too long, better simplify. Also the legend is too extensive; some of the specifications can be insert in the Materials and Methods section.
Author Response
Response to Reviewer 1 comments
Reference#: Animals-585569
Title: Effects of mannan oligosaccharides on gas emission, protein and energy utilization, and fasting metabolism in sheep
We greatly appreciate the comments from the reviewer and have addressed each of these. Our point-to-point responses to the comments are shown in red text in the following pages, and all changes in the revised manuscript have also been indicated in red text.
Point 1: In my opinion the Manuscript titled “Effects of mannan oligosaccharides on gas emission, protein and energy utilization, and fasting metabolism in sheep” (animals-585569) is an original scientific paper in the broad areas of Animals journal. The subject (effects of MOS on in vitro and in vivo gas emission, utilization of crude protein and energy, and relative parameters in sheep under fasting metabolism conditions) is a topic interesting to look into.
Response 1: Thanks a lot for your kind recognition about our works.
Point 2: I think the manuscript needs some revisions before publication.
Response 2: Thanks a lot for your kind comments and advises, and we revised the manuscript totally to avoiding mistakes and unclearness.
Point 3: In general, I think that the manuscript is well written respecting the Animals journal indications.
Response 3: Thanks for your recognition about our works.
Point 4: Keywords: I prefer use words not present in the title
Response 4: Thanks for your comments. But actually, the major key points present in the title, so it is really impossible to choose alternative words to point out the major key points in keywords section.
Point 5: Material and methods: The experimental design is complex due to the in vivo and in vitro part. However, some details should be added, for example amount of medium in the fermentation channel in the in vitro trial, how the conversion of gas for 12h to 24h was made and some specification for the chemical analyses of basal diet (i.e. NDF, ADF, Ca, P reported in table 1). Probably, more references for protocol used needs to be inserted.
Response 5: Thanks for your constructive suggestion. We checked the Materials and Methods section carefully, and revised the unclear parts according to your comments.
Point 6: Results: can be improved (see specific suggestions).
Response 6: Thanks for your constructive suggestion. We revised this part according to your comments to make this part clearer and more logical.
Point 7: Discussion: the results obtained were well discussed and justified; the data have been adequately commented and compared with similar studies.
Response 7: Thanks for your recognition for our works.
Specific suggestions:
Point 8: P1L17: move ‘in vitro’ after ‘led’
Response 8: We have adjusted this sentence.
Point 9: P1L32-33: the sentence is not clear please re-write
Response 9: We have re-written this sentence.
Point 10: P2L72-73: the sentence is not clear please clarify
Response 10: We have clarified this sentence.
Point 11: PL103: normally a flow of CO2 is used…
Response 11: Thank you very much for your advice. 0.2843% CO2 was used to calibrate the gas flowmeter, so we have revised this sentence to make it clearer.
Point 12: P4L134-135: not clear ‘…for energy measurements…’
Response 12: We have revised this sentence.
Point 13: P4L1557-156: change the sentence as follows ‘where X is the observation of the dependent variable (i = 1 to 13), μ is the population mean, α is the random effect of treatment, and e is the random error associated with the observation.’
Response 13: We have revised this sentence according to your comment.
Point 14: P4L157-161: delete this part, it is not necessary
Response 14: We have deleted this part.
Point 15: P4-5L166-174: all this part should be re-written, because it is not easy to understand. The statistical significances are already reported in the table…, you should just underline the highest or lowest values
Response 15: We have re-written this part to make this part clearer.
Point 16: P5L181-182: simplify the title… (also elsewhere)
Response 16: We have revised the titles according to you comment.
Point 17: P8L243: delete ‘approximately’
Response 17: We have deleted this word.
Point 18: P10L324-327: improve the grammar of the sentence (after semicolon there is no verb)
Response 18: We have revised this sentence.
Tables:
Point 19: Table 2: It is not clear why and how this was made ‘Mean results of in vitro gas production are shown for 12-h collection phase of the study for each treatment (n = 3 per treatment), and then converted into yield per 24-h (in vitro gas production (mL/24 h) = average production (mL/min) × 60 × 24).’
Response 19: We have revised this part to make it clearer.
Point 20: In the legend delete a,b before values.
Response 20: We have deleted these characteristics.
Point 21: Tables 3, 4 and 5: The titles are too long, better simplify. Also the legend is too extensive; some of the specifications can be insert in the Materials and Methods section.
Response 21: We have revised the tables according to your comment.
Reviewer 2 Report
The manuscript investigates on the effect of mannan oligosaccarides on gas emission, protein and energy utilization and fasting metabolism in sheep. The results are very interesting. I suggest to improve the description of the in vitro procedures. The authors adopted the Menke method. Did they use syringes? or vessels connected to the flowmeter?
The data of table 2 are very interesting. i suggest to improve the discussion about these results.
Considering the data of table 2, 3 and 4, MOS semms to affect N-metabolism and not methanogenesis. Please, discuss better this point and give an explanation.
Author Response
Response to Reviewer 2 comments
Reference#: Animals-585569
Title: Effects of mannan oligosaccharides on gas emission, protein and energy utilization, and fasting metabolism in sheep
we greatly appreciate the comments from the reviewer and have addressed each of these. Our point-to-point responses to the comments are shown in red text in the following pages, and all changes in the revised manuscript have also been indicated in red text.
Point 1: The manuscript investigates on the effect of mannan oligosaccarides on gas emission, protein and energy utilization and fasting metabolism in sheep. The results are very interesting. I suggest to improve the description of the in vitro procedures. The authors adopted the Menke method. Did they use syringes? or vessels connected to the flowmeter?
Response 1: Thanks a lot for your cognition about our works, and we have improved the in vitro procedures section according to your comments. During the gas emission experiment, we used the automatic recording device of trace gas production with six channels (Branch of Animal Husbandry, Jilin Academy of Agricultural Sciences, Gongzhuling, Jilin Province, P.R.China) equipped with gas flowmeters (CH4 and CO2, Sensors, Inc., Saline, Michigan, USA; H2, CITY Technology, Inc., Hampshire, UK) instead of syringes and vessels connected to the flowmeter.
Point 2: The data of table 2 are very interesting. i suggest to improve the discussion about these results.
Response 2: Thank you very much for your constructive advice. We have revised relative discussion part.
Point 3: Considering the data of table 2, 3 and 4, MOS semms to affect N-metabolism and not methanogenesis. Please, discuss better this point and give an explanation.
Response 3: Thanks for your constructive advices, and we have improved these parts.
Reviewer 3 Report
Line 102: look the instruction for authors about the style of citations
Line 95-175: please use the same format like the title of tab 4 and 5
Line 324: please delete";" after H2 production
Line 347: look the instruction about the format of the reference section
Author Response
Response to Reviewer 3 comments
Reference#: Animals-585569
Title: Effects of mannan oligosaccharides on gas emission, protein and energy utilization, and fasting metabolism in sheep
We greatly appreciate the comments from the reviewer and have addressed each of these. Our point-to-point responses to the comments are shown in red text in the following pages, and all changes in the revised manuscript have also been indicated in red text.
Point 1: Line 102: look the instruction for authors about the style of citations
Response 1: Thanks a lot for your careful check. We have revised this part.
Point 2: Line 95-175: please use the same format like the title of tab 4 and 5
Response 2: Thank you very much for your careful check. We have revised these parts.
Point 3: Line 324: please delete";" after H2 production
Response 3: We have deleted the semicolon.
Point 4: Line 347: look the instruction about the format of the reference section
Response 4: Thank you very much. We have revised references according to the instruction about the format.
Round 2
Reviewer 2 Report
Dear Editor and Authors, the manuscript has been revised and is acceptable in this form